# A PCR plus restriction enzyme-based technique for detecting target-enzyme mutations at position Pro-106 in glyphosate-resistant *Lolium perenne*

Hossein Ghanizadeh[1]☯*, Andrew G. Griffiths[2]‡, Christopher E. Buddenhagen[3]‡, Craig B. Anderson[2]☯, Kerry C. Harrington[1]‡

1 School of Agriculture and Environment, Massey University, Palmerston North, New Zealand,
2 AgResearch Grasslands Research Centre, Palmerston North, New Zealand, 3 AgResearch Limited, Hamilton, Private Bag, New Zealand

☯ These authors contributed equally to this work.
‡ These authors also contributed equally to this work.
* H.Ghanizadeh@massey.ac.nz

**Data Availability Statement:** All relevant data are within the manuscript.

## Abstract

The first step in managing herbicide-resistant weeds is to confirm their resistance status. It is, therefore, crucial to have a rapid, reliable and cost-effective technique to assess samples for herbicide resistance. We designed and evaluated three derived cleaved amplified polymorphic sequence (dCAPS) markers for detecting glyphosate resistance in *Lolium perenne*. conferred by non-synonymous mutations at codon-106 in the *enolpyruvylshikimate-3-phosphate synthase* (*EPSPS*) gene. The dCAPS markers involve amplification of the target region, digestion of the amplified products with restriction enzymes and gel-based visualisation of the digested products. The results showed that all three dCAPS markers could successfully detect mutations at codon-106 in the target enzyme. The dCAPS markers can also inform us of the zygosity state of the resistance allele and was confirmed by sequencing the target region of the *EPSPS* gene. The markers described here are effective quick tests for the monitoring and evaluation of the target-enzyme mechanism of glyphosate resistance in *Lolium perenne*.

## Introduction

Herbicides are valuable tools for weed control and crop production; however, their usefulness is being undermined by the development of herbicide resistance [1, 2]. Currently there are over 500 unique cases of herbicide-resistant weeds globally [3], and this is a growing issue compounded by the lack of new herbicide modes of action being deployed in recent years [4, 5]. Recent studies have characterized chemical classes with the potential to be developed into herbicides with novel modes of action [5]. However, synthesizing and marketing a new mode of action is an expensive and time-consuming process [4]. A sustainable weed management approach, therefore, needs to be implemented to maximize the efficacy of the herbicides currently available on the market [6, 7].

**Funding:** Financial support for the authors was provided by the Endeavour fund (C10X1806, Managing Herbicide Resistance) from the New Zealand Ministry for Business Innovation and Employment. The funder did not play any role in the study design, data collection and analysis, decision to publish, or preparation of the manuscript.

**Competing interests:** The authors have declared that no competing interests exist.

The first step towards managing suspected herbicide resistance is to detect that resistance has developed [8]. The conventional method for evaluating herbicide resistance is whole-plant dose-response assays [8], which is a time-consuming process (1–3 months). An alternative approach to this method is to use quick tests [9–12]. Quick tests are more rapid and less labour-intensive than conventional methods for detecting herbicide resistance [12]. Depending on the type of quick test, a variety of plant material, including seed, leaf tissue or DNA samples, can be used [8]. In the case of glyphosate resistance, quick tests such as seed-based assays and *in vitro* enzymatic assays have been developed for detecting resistance in weeds [13, 14]. By contrast, quick tests focussing on the specific DNA mutations (i.e. molecular markers) responsible for resistance in phenotypes have seldom been used to evaluate glyphosate resistance in weeds. Implementing such a method requires elucidation of the resistance mechanism at the DNA sequence level, enabling development of quick tests such as PCR (polymerase chain reaction)-based assays to identify cases of resistance [8]. As opposed to methods that require sequencing of the target regions, these rapid PCR-based assays most commonly use restriction fragment length polymorphism (RFLP) techniques to detect the causative single nucleotide polymorphisms (SNPs) in regions of the target enzymes to which herbicides bind [15]. Cleaved amplified polymorphic sequence (CAPS) and derived cleaved amplified polymorphic sequence (dCAPS) are two common RFLP assays using restriction enzymes to cleave PCR amplicons to detect SNPs in herbicide-resistant weeds [15]. The dCAPS assay is a modified version of the CAPS [16]. In the CAPS assays, restriction enzyme recognition and cleavage sites diagnostic of the resistance SNP mutation are present in the target sequence, but this method is limited by whether the mutation falls within the recognition sites for available restriction enzymes. By contrast, in dCAPS assays additional SNPs are incorporated in the PCR primers to align with the target SNPs to create diagnostic restriction enzyme cleavage sites [16]. This methodology allows greater versatility for designing target site mutation quick tests as creating restriction enzyme cleavage sites with the target SNPs increases the pool of available enzymes that may be used for diagnostic assays [15].

Glyphosate, a widely used herbicide, has shown development of resistance across a wide range of species [3]. It is non-selective, systemic and controls weed species in agricultural and non-agricultural scenarios [17]. The popularity of this herbicide is due to it being inexpensive, very effective and has no soil-residual activity [17–19]. These properties have led to glyphosate being commonly used to control weeds in many diverse environments such as orchards, vineyards, railways, roadsides, and urban areas [19]. In the late 1990s, the first case of resistance to glyphosate in the world was reported from Australia in *Lolium rigidum* [20]. Since then, glyphosate resistance in other weed species has been identified, including other *Lolium* species, *L. multiflorum* and *L. perenne* [21–23]. Glyphosate resistance in *Lolium* spp., is an issue of global importance, particularly for perennial ryegrass (*L. perenne*), an agronomically significant forage species underpinning temperate agriculture that is often subjected to glyphosate pressure [6].

Glyphosate resistance can be due to non-target-site or target-site mechanisms of resistance [24, 25]. The target-site mechanism of resistance includes target-enzyme mutation and target-enzyme over-expression [25]. To date, three different combinations of the target-enzyme mutations (namely, single, double and triple amino acid substitutions) have been identified that confer resistance to glyphosate in a wide range (>25) of species [3, 26]. The single non-synonymous substitution involves a mutation in the first and second nucleobases of the triplets replacing the glyphosate-susceptible amino acid Pro at codon-106 in the *EPSPS* gene [25]. Additional amino acid substitutions at codon positions 102 and 103 also enhance glyphosate resistance, but only in combination with substitution of Pro-106 [26]. This codon provides a conserved target for developing a quick PCR-based test to screen suspected incidences of glyphosate resistance caused by the Pro-106 mutation.

**Table 1. dCAPS primers designed to detect target-enzyme mutation at position Pro-106 in *EPSPS* gene in *Lolium* perenne.**

| Primers[a] | Restriction enzymes[b] | Predicted amplicon size (bp) | Digested fragment Size (bp) |
|---|---|---|---|
| F1:5`-TAAAGCTCTTCCTGGGGAACGCTGGAACTGCGATG<u>T</u>GG-3` | *Msc*I (TGG/CCA) | 216 | 178 + 38 |
| R1:5`- GGTCGCTCCCTCATTCTTGGTACTCCATCAAGAA-3` | | | |
| F2:5`- TAAAGCTCTTCCTGGGGAACGCTGGAACTGCGATG<u>G</u>GG-3` | *Sau*96I (G/GNCC) | 216 | 180 + 36 |
| R2:5`- GGTCGCTCCCTCATTCTTGGTACTCCATCAAGAA-3` | | | |
| F3:5`- TAAAGCTCTTCCTGGGGAACGCTGGAACTGCGATG<u>GG</u>G-3` | *Nla*IV (GGN/NCC) | 216 | 179 + 37 |
| R3:5`- GGTCGCTCCCTCATTCTTGGTACTCCATCAAGAA-3` | | | |

[a]The introduced mismatch in the designed dCAPS primer is underlined.

[b]The restriction enzyme recognition sequence is in brackets; / = cut site; N = any nucleotide.

The aim of this research was to design, optimise and evaluate the suitability of a dCAPS marker system for rapid detection of mutations at the position Pro-106 associated with glyphosate resistance in globally and agronomically important *Lolium* perenne.

## Materials and methods

### Plant material and DNA extraction for developing the assay

Two populations of perennial ryegrass (*Lolium perenne*) were used in this part of the research: a glyphosate-resistant population (population O) with a known target mutation at the position Pro-106 [27] and a known susceptible population (population SP) [28]. Genomic DNA from populations O and SP was extracted and quantified as described previously [29].

### dCAPS primers and protocol development

Three different dCAPS primer pairs specific to each of three restriction enzymes (*Msc*I (TCC/CCA); *Sau*96I (G/GNCC); *Nla*IV (GGN/NCC)) were designed using a draft genome sequence of perennial ryegrass (AgResearch Ltd.) to evaluate SNPs at the position Pro-106 in the *EPSPS* gene (Tables 1 and 2). For restriction enzyme specificity, the forward primer annealed immediately 3' of the codon 106 and was modified by introducing a mismatch at the N-3 position of this codon. This was to create a restriction site for the corresponding restriction enzyme in the absence of the codon 106 mutation that confers glyphosate resistance. These primers were predicted to generate a 216 bp polymerase chain reaction (PCR) amplicon that would yield 178, 180 and 179 bp fragments after digestion with *Msc*I, *Sau*96I, and *Nla*IV, respectively, in the presence of the wild-type material sensitive to glyphosate at codon 106 (Pro-106). The 216 bp amplicon would not be cleaved by these enzymes if it contained the resistance mutation. The optimum annealing temperature for the designed dCAPS primers was evaluated using a gradient annealing temperature and magnesium concentration (1.5–3.0 mM) approach [30]. The

**Table 2. Nucleotide sequence in *EPSPS* gene in glyphosate-resistant (O) and glyphosate-susceptible (SP) populations of perennial ryegrass.** Changes to codons from the consensus sequence are highlighted.

| Codon number | 100 | 101 | 102 | 103 | 104 | 105 | 106 | 107 | 108 |
|---|---|---|---|---|---|---|---|---|---|
| Amino acid | Ala | Gly | Thr | Ala | Met | Arg | Pro | Leu | Thr |
| Consensus sequence | GCT | GGA | ACT | GCG | ATG | CGG | CCA | TTG | ACG |
| SP | GCT | GGA | ACT | GCG | ATG | CGG | CCA | TTG | ACG |
| O | GCT | GGA | ACT | GCG | ATG | CGG | *T/CCA | TTG | ACG |

* The resistant individuals investigated here were heterozygous at this position.

PCR amplification reaction contained template perennial ryegrass genomic DNA (~20–30 ng), dNTP (200 μM) (NEB, UK), forward and reverse primers (each 0.2 μM), MgCl$_2$ (0.5mM), *Taq* DNA Polymerase (1.25 units) (NEB, UK), *Taq* DNA Polymerase buffer (2.5 μl) (NEB, UK) and nuclease-free water to bring the volume of reaction to 25 μl. The PCR cycling program included initial denaturation at 95˚C (one cycle of 2 min), denaturation at 95˚C (40 cycles of 30 s), 40 cycles of 30 s annealing at 60˚C (for all three primer pairs), 40 cycles of 30 s extension at 68˚C, followed by one cycle final extension at 68˚C (10 min). The PCR products were then used directly for restriction digestions. The restriction digestion reaction (10 μl) contained 8.5 μl of PCR products, 1 μl of the appropriate buffer and 0.5 μl (5 units) of *Msc*I and *Sau*96I enzymes (NEB, UK) for Gly-dCAPS-F1/ Gly-dCAPS-R1 and Gly-dCAPS-F2/ Gly-dCAPS-R2 primers, respectively. The restriction digestion reaction (10μl) for Gly-dCAPS-F3/ Gly-dCAPS-R3 primers contained 7.75 μl of PCR products, 1 μl of buffer and 1.25 μl (2.5 units) of *Nla*IV enzyme (NEB, UK). The reactions were incubated at 37˚C for 1.5 h, and the digested products were loaded on an agarose 1x LB (lithium borate) 2% (w v$^{-1}$) gel containing 0.5 μg ml$^{-1}$ ethidium bromide before they were run at 4 V cm$^{-1}$ for 1.5 h and visualised under UV illumination using a Gel Doc XR 2000 system (Bio-Rad Laboratories).

### Confirmation of dCAPS markers using EPSPS genotyping

To provide sequence confirmation of dCAPS marker results, a pair of primers (Glyf1-CGTG-GAAGCAGACAAAGTTGC/Glyr1-GGTCGCTCCCTCATTCTTG) to PCR amplify a 300 bp portion of the perennial ryegrass *EPSPS* gene flanking codon 106 were designed using a draft genome sequence of perennial ryegrass (AgResearch Ltd.). PCR amplification and sequencing of this *EPSPS* region was carried out as follows. The PCR reaction contained template perennial ryegrass genomic DNA (~10 ng), dNTP (200 μM) (NEB, UK), forward and reverse primers (each 0.2 μM), *Taq* DNA Polymerase (1.25 units) (NEB, UK), *Taq* DNA Polymerase buffer (2.5 μl) (NEB, UK) and nuclease-free water to bring the volume of reaction to 25 μl. The PCR cycling program included initial denaturation at 95˚C (one cycle of 2 min), denaturation at 95˚C (35 cycles of 30 s), 35 cycles of 30 s annealing at 58˚C, 35 cycles of 30 s extension at 68˚C, followed by one cycle final extension at 68˚C (10 min). The PCR products were then evaluated by resolution and visualisation on a 1x LB 1% (w v$^{-1}$) agarose gel containing 0.5 μg ml$^{-1}$ ethidium prior to sequencing at the Massey University Genomic Centre (https://www.massey.ac.nz/massey/learning/departments/centres-research/genome/massey-genome-service-home.cfm). The sequence data were evaluated to confirm the presence of a point mutation at codon 106 in population O using the method described previously [31].

### *Lolium perenne* test population to assess the dCAPS assay

A perennial ryegrass population exhibiting glyphosate resistance was selected for evaluation using the *EPSPS* target-site mutation dCAPS assay. This population was developed by collecting four individual ryegrass plants exhibiting glyphosate resistance from a vineyard in Blenheim, New Zealand. Preliminary evaluation of the four plants using the method described previously [28], confirmed their resistance to glyphosate. These plants were allowed to cross-pollinate freely among themselves in an isolation glasshouse under conditions described previously [32], and seeds were collected from each mother plant at maturity. The collected seeds were bulked and labelled as 'Blenheim' population. A random sample of 144 seeds from this Blenheim population was planted into 10 cm by 10 cm plastic pots containing a potting medium (40% bark fibre, 20% C.A.N. fines A Grade, 20% coco fibre, and 20% pumice) with slow release fertiliser, and the pots were kept in a glasshouse with an average temperature of 19˚C and a relative humidity of 80% under natural light. The resulting 144 individual plants

were grown sufficiently large enough for each plant to be separated into eight individual tillers (clones). These clones were grown until they each had 3–4 leaves before being sprayed with glyphosate (Weedmaster 540 g ae L$^{-1}$) using a dual-nozzle laboratory track sprayer calibrated to deliver 220 L ha$^{-1}$ at 200 kPa with 0.1% organosilicone adjuvant Pulse. Each tiller was subject to one treatment of glyphosate at either 0, 100, 360, 540, 860 or 2000 g ae ha$^{-1}$. Survival of clones at each glyphosate rate was assessed at 4 weeks after glyphosate treatment. The clones (n = 34) that survived the glyphosate rate of 2000 g ha$^{-1}$ were poly-crossed to generate a F$_2$ generation expected to comprise individuals homozygous or heterozygous for the glyphosate resistance/sensitive *EPSPS* codon 106 alleles. From this F$_2$ population, 18 randomly selected progenies were assessed for the glyphosate target-enzyme resistance mechanism using the dCAPS assay. For this, tissue samples (leaf segments) were collected from individual plants and genomic DNA was extracted from samples using the method described by Anderson, Franzmayr [33]. The same dCAPS procedure as described above was performed using only the *Sau*96I enzyme and corresponding primers.

## Results

### Evaluating target-enzyme SNPs for populations O and SP

The results from the dCAPS primers evaluating the target-enzyme mutation at the codon 106 of *EPSPS* gene in the glyphosate-susceptible (SP) and glyphosate-resistant (O) perennial ryegrass populations are illustrated in Fig 1. All three dCAPS primer pairs generated the predicted 216 bp fragment after PCR amplification from samples of populations SP and O. Following

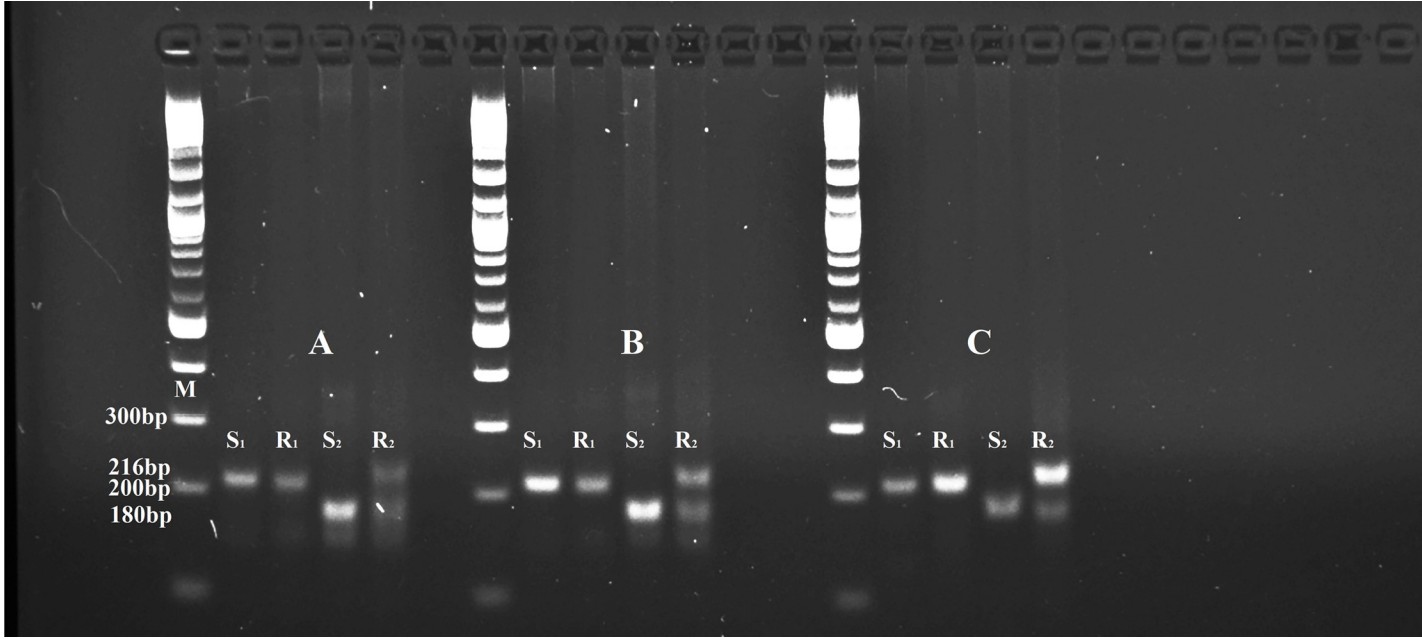

**Fig 1.** Digestion patterns of three restriction enzymes, *Msc*I (A), *Sau*96I (B) and *Nla*IV (C) used for the dCAPS markers for detecting mutations at position Pro-106 in *EPSPS* gene in glyphosate-resistant population O (R) and glyphosate-susceptible population SP (S). S$_1$ and R$_1$ represent undigested samples showing the 216 bp dCAPS amplicon from an individual each of population S and O, respectively. S$_2$ represents restriction enzyme-digested samples for the glyphosate susceptible individual yielding a single ~180 bp fragment. This individual was homozygous for the wild-type glyphosate sensitive allele of the *EPSPS* gene that harbors the C nucleotide at the codon 106 required for restriction enzyme cleavage in these dCAPS assays. The glyphosate-resistant individual (R$_2$) had a 216 bp and ~180 bp fragment showing the presence of a mutation at codon 106 of the *EPSPS* gene for one allele as well as the wild-type sensitive allele, indicating this plant was heterozygous for these alleles. M denotes the DNA size standard lanes showing the 200 and 300 bp fragments of the 1 kb Plus size marker ladder (NEB, UK). The samples were resolved and visualized by electrophoresis in an agarose lithium borate buffer (2% w v$^{-1}$) gel containing 0.5 μg ml$^{-1}$ ethidium bromide.

digestion with the corresponding restriction enzymes, the amplicons generated from the glyphosate-susceptible population SP were cleaved to yield a ~180 bp fragment, as predicted, indicating that this individual was homozygous for the wild-type glyphosate-sensitive allele (Fig 1). By contrast, the amplicons generated from the glyphosate-resistant O population individual yielded both a cleaved wild-type glyphosate-sensitive allele ~180 bp fragment and an uncut 216 bp glyphosate resistance allele where there was no restriction enzyme recognition site in the presence of the target-site resistance SNP mutation at codon 106 (Fig 1). This indicated this individual was heterozygous for the glyphosate resistance allele.

Sequence confirmation of the dCAPS results from a resistant individual of population O was provided by partial sequencing of the *EPSPS* gene using Glyf1/Glyr1 primers. These primers generated the 300 bp DNA amplicon containing the glyphosate resistance mutation site at codon 106 (Table 2). Amplicon sequencing identified both a C and a T nucleotide at codon 106 as revealed by a double nucleotide peak in the corresponding position in the sequencing chromatogram (Fig 2). The mutation converts proline (Pro) to serine (Ser) at amino acid 106 in the population O, which confers glyphosate resistance. These sequence data confirmed not only the presence of a C to T substitution mutation at this codon, but also the heterozygous glyphosate resistant/susceptible allele state revealed by the dCAPS assay (Fig 1).

### Evaluating target-enzyme SNPs within the Blenheim population

The dCAPS target-site mutation detection assay was deployed in a $F_2$ perennial ryegrass population that had shown resistance to glyphosate. Since we obtained similar results for *Msc*I, *Sau*96I and *Nla*IV enzymes and all three restriction enzymes were able to detect SNP variation at codon 106 of the *EPSPS* gene, we chose to use only one enzyme for assessing target-enzyme mutations at this position. The selected restriction enzyme was *Sau*96I due to its greater cost-effectiveness compared to the other two enzyme options. Partial sequencing of the *EPSPS* gene of the mother plants from which the $F_2$ individuals were collected using Glyf1/Glyr1 primers revealed that a proline to serine substitution at the codon 106 of the *EPSPS* gene was present in all mother plants (n = 34), and the mother plants were heterozygous for this point mutation as confirmed by sequence results (S1 Fig). The dCAPS results of the $F_2$ progeny showed that all 18 individuals from the Blenheim population contained a SNP at codon 106 that prevented cleavage by *Sau*96I showing loss of the susceptible allele (Fig 3). Additionally, the majority [16] also had the cleaved (~180 bp) dCAPS amplicon of the glyphosate-sensitive allele, indicating these individuals were heterozygous for the resistant/sensitive alleles (Fig 3). As expected with segregation within this population, a subset of two out of the 18 $F_2$ individual plants were found to be homozygous for the glyphosate resistant dCAPS amplicon at codon 106 of the *EPSPS* gene. Similarly, amplicon sequencing of both homozygous individuals only identified a T nucleotide at codon 106 as revealed by a single nucleotide peak in the corresponding position in the sequencing chromatogram (Fig 4).

### Discussion

Both Italian ryegrass and perennial ryegrass are important pasture species [6]; however, both species are problematic weeds in places such as vineyards and arable crops [23]. Evolution of resistance to 14 different herbicide modes of action has been confirmed for *Lolium* perenne. globally [3]. Given the limited chemical options available for weed management in vineyards [4], it is crucial to take a proactive approach to herbicide resistance management.

Successful management of herbicide resistance requires quick and accurate methods for detecting herbicide-resistant weeds. Previously, we evaluated three potential quick tests, namely, seed-based assay, shikimic acid assay and tiller dip assay for detecting glyphosate

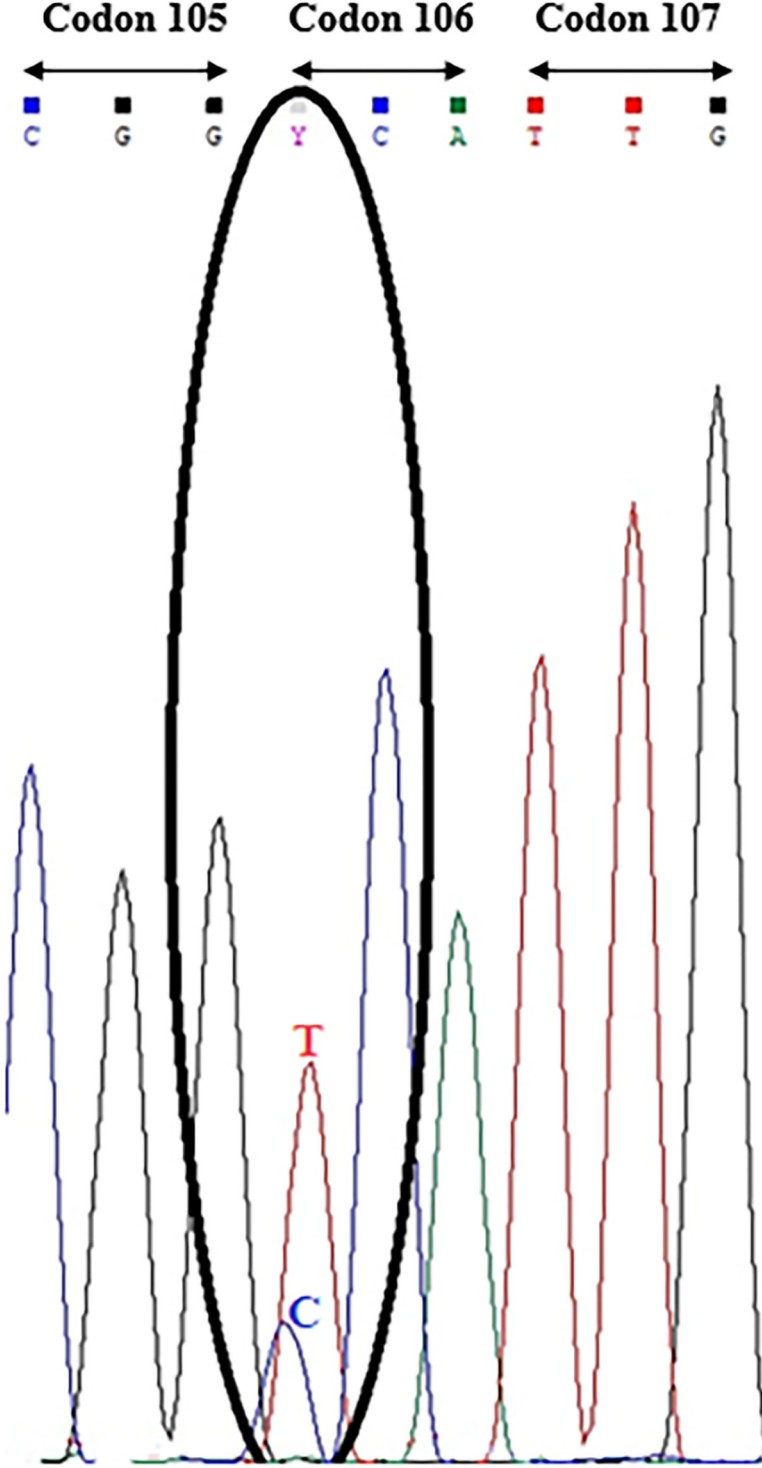

**Fig 2. Sequence chromatogram showing codons 105 to 107 of a 300 bp PCR amplified section of the *EPSPS* gene of a perennial ryegrass individual from glyphosate resistant population O.** As both alleles in the individual were sequenced simultaneously the chromatogram revealed a heterozygous state at the first base of codon 106 (circled) identified as a pyrimidine (Y) comprising the wild type glyphosate sensitive C nucleotide overlapping with the glyphosate resistant T mutation at this position. This non-synonymous mutation indicated one allele in this glyphosate resistant individual had a Pro (CCA) to Ser (TCA) substitution at codon 106.

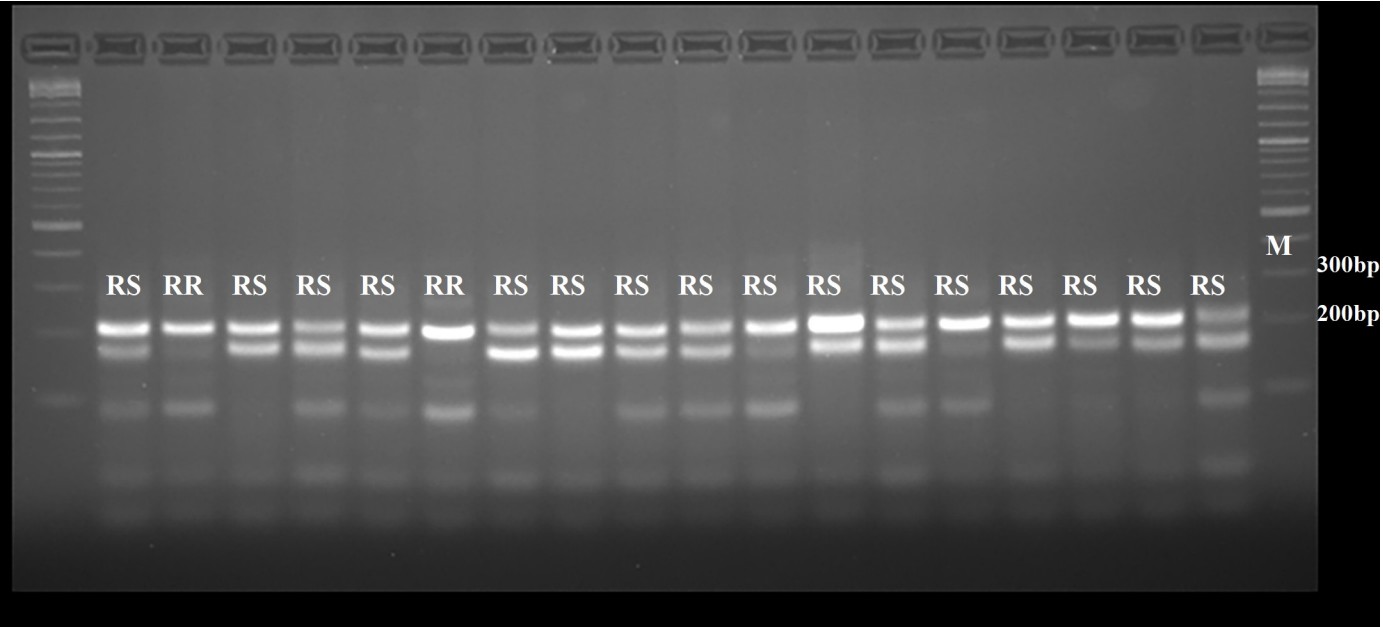

**Fig 3. Digestion patterns of the restriction enzyme, *Sau*96I used for the dCAPS markers for detecting mutations at codon 106 in the *EPSPS* gene in 18 individuals from a glyphosate-resistant Blenheim population.** The presence of the 216 bp dCAPS amplicon indicated a non-cleaved resistant allele of the *EPSPS* gene, whereas the ~180 bp cleaved amplicon showed presence of the glyphosate-sensitive allele. RS and RR represent digested samples for glyphosate-resistant individuals that were heterozygous and homozygous, for SNP mutations conferring resistance at codon 106 in the *EPSPS* gene. M denotes the DNA size standard lanes showing the 200 and 300 bp fragments of the 1 kb Plus size marker ladder (NEB, UK). The samples were resolved and visualized by electrophoresis in an agarose lithium borate buffer (2% w v$^{-1}$) gel containing 0.5 μg ml$^{-1}$ ethidium bromide.

resistance in ryegrass [13]. In this research, we evaluated a dCAPS marker based on a well-known target-site mutation for detecting glyphosate resistance in ryegrass. dCAPS markers have mostly been used to detect target-enzyme mutations conferring resistance to acetyl-CoA carboxylase (ACCase)- and acetolactate synthase (ALS)-inhibitors [8]. dCAPS markers have also been used for detecting target-enzyme mutations in glyphosate-resistant *Amaranthus tuberculatus* [34] and *Eleusine indica* [35]. However, to the best of our knowledge, no dCAPS markers have been developed before that specifically detect target-enzyme mutations in glyphosate-resistant ryegrass.

Compared to other mechanisms that confer glyphosate resistance, there are fewer cases in which target-enzyme mutations have been reported to cause glyphosate resistance in weed species [25]. Depending on the nature of the target-enzyme mutation and the number of combinations of amino acid substitutions (i.e. single, double and triple mutations), different levels of resistance have been documented for glyphosate-resistant weeds with the target-enzyme mutation mechanism [26]. The mutations at the position Pro-106 can confer low to moderate ($<$ 6-fold) levels of resistance to glyphosate [25]; however, in combination with other positions (e.g. Thr-102), amino acid substitutions at the position Pro-106 can cause a high-level glyphosate resistance [25, 26, 35]. Up to five amino acid substitutions at the position Pro-106 have been documented [25]. At codon 106, proline is coded by the CCN triplet thus any changes at the first and second nucleobases of the triplets can result in amino acid replacements at this codon. The restriction enzymes used in the dCAPS technique developed in this research can detect the wild type proline, hence any nucleotide changes at the first or second bases of the triplet at codon 106 can disrupt the restriction enzyme recognition site and result in uncut bands. Thus, the dCAPS markers studied here enable researchers to confirm the presence of absence of resistance alleles at the position Pro-106 regardless of the nature of the alleles.

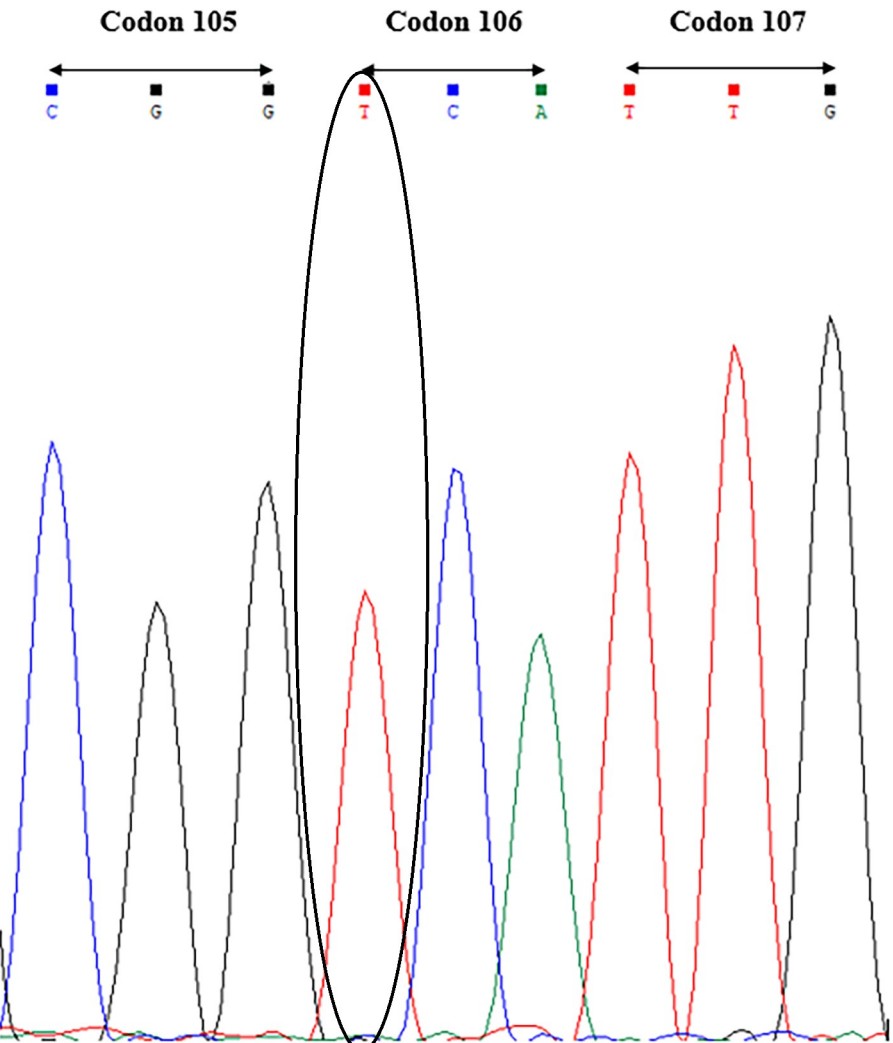

**Fig 4. Sequence chromatogram showing codons 105 to 107 of a 300 bp PCR amplified section of the *EPSPS* gene of a perennial ryegrass individual from Blenheim population.** As there were two resistance alleles at position 106, the chromatogram revealed a homozygous state at the first base of codon 106 (circled) identified as the glyphosate resistant T mutation at this position. This non-synonymous mutation indicated both alleles in this glyphosate resistant individual had a Pro (CCA) to Ser (TCA) substitution at codon 106.

DNA-based assays offer several advantages over other methodologies. The DNA-based assays compared to methods such as seed-based assays are not tissue specific as DNA can be extracted from all types of plant tissues. Also, DNA-based assays are rapid, and results can be achieved within 8–48 hours after sampling. In addition to resistance confirmation, DNA-based assays can tell us if the target-enzyme mutations are involved in the mechanism of resistance. The dCAPS marker has the added benefit of allowing us to detect whether the plant is homozygous or heterozygous for the mutation. The knowledge of the zygosity state of resistance alleles has a crucial role in the level of resistance that is conferred by target-enzyme mutations to herbicides such glyphosate [25, 32, 36, 37]. However, the limitation to the DNA-based assays is that these assays are only applicable to resistance mechanisms for which the sequence of the target-enzyme is known [8]. DNA-based assays such as dCAPS do not provide information about the type of mutation that disrupts the recognition site of restriction

enzymes [15]. Also, this assay may need to be optimised for different weed species given the nucleotide sequences are relatively variable among different species [8].

PCR plus restriction enzyme-based assays can be useful tools for detecting resistance to herbicides in large-scale surveys when evaluating the distribution of herbicide-resistant weeds [8]. In this case, molecular markers such as dCAPS markers will be useful in order to get a preliminary evaluation on samples before confirmation with other means. For instance, if a suspected glyphosate-resistant population tested positive for the target-site mechanism of resistance, then the population can be labelled as herbicide-resistant if the objective is to provide a yes/no confirmation of resistance.

## Conclusion

The PCR plus restriction enzyme-based assay developed in this study can be used to evaluate the target-enzyme mechanism of glyphosate resistance rapidly and shed light on the molecular basis for resistance to glyphosate in *Lolium* perenne. Such PCR plus restriction enzyme-based tools would enable an analysis of large numbers of samples in a short time; thus, future studies will involve assessing across a wide range of glyphosate-resistance and susceptible ryegrass material using the dCAPS assay developed in this research.

## Supporting information

**S1 Fig. Partial DNA sequencing of *EPSPS* gene in 34 heterozygous glyphosate-resistant plants.** The arrow shows the SNP mutation conferring resistance at codon 106 in the *EPSPS* gene. As both alleles in the individual were sequenced simultaneously a heterozygous state at the first base of codon 106 (arrow) identified as a pyrimidine (Y) comprising the wild type glyphosate sensitive C nucleotide and the glyphosate-resistant T mutation at this position. (TIF)

**S1 Raw images.**
(PDF)

## Acknowledgments

We thank the editor and the anonymous reviewer for their constructive comments, which helped us to improve the manuscript.

## Author Contributions

**Conceptualization:** Hossein Ghanizadeh, Andrew G. Griffiths, Craig B. Anderson.

**Data curation:** Hossein Ghanizadeh.

**Formal analysis:** Hossein Ghanizadeh.

**Funding acquisition:** Hossein Ghanizadeh, Andrew G. Griffiths.

**Investigation:** Hossein Ghanizadeh.

**Methodology:** Hossein Ghanizadeh, Andrew G. Griffiths, Christopher E. Buddenhagen.

**Project administration:** Andrew G. Griffiths, Kerry C. Harrington.

**Resources:** Andrew G. Griffiths, Christopher E. Buddenhagen, Kerry C. Harrington.

**Software:** Hossein Ghanizadeh.

**Validation:** Hossein Ghanizadeh.

**Visualization:** Hossein Ghanizadeh, Craig B. Anderson.

**Writing – original draft:** Hossein Ghanizadeh.

**Writing – review & editing:** Andrew G. Griffiths, Christopher E. Buddenhagen, Craig B. Anderson, Kerry C. Harrington.

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
