## [Decision Letter · Decision Letter 0]

5 Jan 2021

PONE-D-20-36361

A PCR plus restriction enzyme-based technique for detecting target-enzyme mutations at position Pro-106 in glyphosate-resistant Lolium perenne

PLOS ONE

Dear Dr. Ghanizadeh,

Thank you for submitting your manuscript to PLOS ONE. After careful consideration, we feel that it has merit but does not fully meet PLOS ONE’s publication criteria as it currently stands. Therefore, we invite you to submit a revised version of the manuscript that addresses the points raised during the review process.

I have difficulties to get the second reviewer, and I think the review comments from the sole reviewer are fair. Please revise accordingly.

We look forward to receiving your revised manuscript.

Kind regards,

David D Fang, Ph.D.

Academic Editor

PLOS ONE

Journal Requirements:

4. Please include a caption for figure 4.

5. Please ensure that you refer to Figure 4 in your text as, if accepted, production will need this reference to link the reader to the figure.

Reviewers' comments:

Reviewer's Responses to Questions

**Comments to the Author**

1. Is the manuscript technically sound, and do the data support the conclusions?

Reviewer #1: Yes

2. Has the statistical analysis been performed appropriately and rigorously? 

Reviewer #1: N/A

3. Have the authors made all data underlying the findings in their manuscript fully available?

Reviewer #1: Yes

4. Is the manuscript presented in an intelligible fashion and written in standard English?

Reviewer #1: Yes

5. Review Comments to the Author

Reviewer #1: Comments:

Manuscript # PONE-D-20-36361

The manuscript by Ghanizadeh et at. describes the technique of detecting mutations on EPSPS at the position Pro-106 for perennial ryegrass (Lolium perenne), which was developed based on the dCAPS method published earlier by Neff et al. (Plant Journal 1998). The primers were designed in a way that the PCR products can only be cut by the enzymes when the codon coding for proline-106 at the first and second nucleobases are not mutated which is susceptible to glyphosate. These dCAPS target-site mutation markers were further tested using progeny from the ‘Blenheim’ population. This approach/technique can detect the target enzyme mutations as well as the zygosity of the mutant alleles. The manuscript generally is well written, and the experimental design is fine, and the experiments were apparently well-executed. However, I have a few comments to the manuscript, listed below.

In table 1, the MscI cut site should be TGG/CCA.

Legends in figures 1 and 3, lines 446 and 467, the term “company” should be specified.

In figure 3, “RR” glyphosate-resistant individuals (lanes 11 and 14) seem to have the cleaved (~180 bp) dCAPS amplicon, although the bands are faint. The authors may want to elaborate it a bit in the text.

There are 4 figures in the manuscript. Figure 4 (chromatogram) did not have legend and was not mentioned in the text.

6. PLOS authors have the option to publish the peer review history of their article (what does this mean?). If published, this will include your full peer review and any attached files.

Reviewer #1: No

---

## [Author Response · Author response to Decision Letter 0]

7 Jan 2021

All the modifications have been indicated using the track changes feature in MS Word. Detailed responses to the reviewers are:

Editorial requirements

1- Please ensure that your manuscript meets PLOS ONE's style requirements, including those for file naming.

Response: We have formatted our manuscript according to PLOS ONE`s style requirements.

2- We note that you have included the phrase “data not shown” in your manuscript. Unfortunately, this does not meet our data sharing requirements. PLOS does not permit references to inaccessible data. We require that authors provide all relevant data within the paper, Supporting Information files, or in an acceptable, public repository. Please add a citation to support this phrase or upload the data that corresponds with these findings to a stable repository (such as Figshare or Dryad) and provide and URLs, DOIs, or accession numbers that may be used to access these data. Or, if the data are not a core part of the research being presented in your study, we ask that you remove the phrase that refers to these data.

Response: The authors agreed to submit the partial sequence of EPSPS gene as a supplementary file (S1 Fig) as we believe it would be more convenient for the readers to access the data when they are available as supplementary materials. 

Response: The authors agreed to submit the original uncropped figures. Thus, we provided the original images of gels for the revised version of manuscript.

4. Please include a caption for figure 4.

Response: A caption has been included for Fig 4.

5. Please ensure that you refer to Figure 4 in your text as, if accepted, production will need this reference to link the reader to the figure.

Response: A sentence was added (Lines 272-275) referring to Fig 4.

Reviewer

Firstly, we would like to thank this reviewer for their comments. This reviewer requested a few modifications and they were addressed as below:

In table 1, the MscI cut site should be TGG/CCA.

Response: Thanks for spotting this typo. This has been modified. 

Legends in figures 1 and 3, lines 446 and 467, the term “company” should be specified.

Response: The term has been specified. The name of the company where the ladder was purchased was added.

In figure 3, “RR” glyphosate-resistant individuals (lanes 11 and 14) seem to have the cleaved (~180 bp) dCAPS amplicon, although the bands are faint. The authors may want to elaborate it a bit in the text.

Response: Thanks for spotting this typo in the figure. This has been modified.

There are 4 figures in the manuscript. Figure 4 (chromatogram) did not have legend and was not mentioned in the text.

Response: A caption has been included for Fig 4 and a sentence was added (Lines 272-275) referring to Fig 4.

---

## [Editor Report · Decision Letter 1]

13 Jan 2021

A PCR plus restriction enzyme-based technique for detecting target-enzyme mutations at position Pro-106 in glyphosate-resistant Lolium perenne.

PONE-D-20-36361R1

Dear Dr. Ghanizadeh,

We’re pleased to inform you that your manuscript has been judged scientifically suitable for publication and will be formally accepted for publication once it meets all outstanding technical requirements.

Kind regards,

David D Fang, Ph.D.

Academic Editor

PLOS ONE
---

## [Editor Report · Acceptance letter]

22 Jan 2021

PONE-D-20-36361R1 

A PCR plus restriction enzyme-based technique for detecting target-enzyme mutations at position Pro-106 in glyphosate-resistant *Lolium perenne*. 

Dear Dr. Ghanizadeh:

I'm pleased to inform you that your manuscript has been deemed suitable for publication in PLOS ONE. Congratulations! Your manuscript is now with our production department. 

Kind regards, 

on behalf of

Dr. David D Fang 

Academic Editor

PLOS ONE